# A Study on the Fatigue Performance and Corrosion Resistance of 304/45 Bimetallic Composite Bolts

**DOI:** 10.3390/ma16124454

**Published:** 2023-06-18

**Authors:** Ziming Zhou, Yi Ding

**Affiliations:** College of Materials Science and Engineering, Nanjing Tech University, Nanjing 210009, China; 202061203217@njtech.edu.cn

**Keywords:** composite bolt, fatigue performance, rotary bending, corrosion resistance, microstructure

## Abstract

This paper utilized a hot-rolling process to produce composite rods and subsequently manufactured 304/45 composite bolts through the process of drawing and thread rolling. The study focused on examining the microstructure, fatigue performance, and corrosion resistance of these composite bolts. Additionally, the impacts of quenching and tempering on the fatigue performance of the composite bolts were explored and compared to the performance of 304 stainless steel (SS) bolts and Grade 6.8 35K carbon steel (CS) bolts. The results indicate that the SS cladding of the cold-worked 304/45 composite (304/45-CW) bolts was primarily strengthened by the cold deformation mechanism, which resulted in high microhardness, averaging 474 HV. At a maximum surface bending stress of 300 MPa, the fatigue cycles of the 304/45-CW reached 342,600 cycles at a 63.2% failure probability, which was significantly higher than that of commercial 35K CS bolts. The S-N fatigue curves showed that the fatigue strength of the 304/45-CW bolts was approximately 240 MPa, but the fatigue strength of the quenched and tempered 304/45 composite (304/45-QT) bolts decreased significantly to 85 MPa, due to the loss of the cold deformation strengthening effect. The corrosion resistance of the SS cladding of the 304/45-CW bolts was impressive and remained largely unaffected by carbon element diffusion.

## 1. Introduction

High-strength bolts have the characteristics of a simple installation, stable quality, high construction efficiency, and convenient disassembly. They have become the main connection method for steel structures [1,2,3,4]. However, when they are applied to extremely harsh environments, such as cross-sea bridges, offshore platforms, offshore wind power, and petrochemicals, they will face the risk of corrosion and damage [5,6,7]. Although carbon steel (CS) or low-alloy steel bolts have a high strength, their corrosion resistance is relatively poor [8]. Although SS bolts have a good corrosion resistance, their strength is low and price is high [9,10]. Therefore, how to tightly combine these two metal materials with different physical and chemical properties through mechanical or metallurgical methods, which can meet the requirements of a high strength, high fatigue strength, and good corrosion resistance, is a key focus of current research on marine engineering materials.

Bimetallic composite material is an advanced composite technology that closely combines two different metal materials to form a new metal structure [11,12]. It not only has the superior performance of the two metals, but also has a complementary effect that improves the comprehensive performance of the metal, thereby enhancing its application value [13,14]. With a wide range of applications, bimetallic composite material can meet the needs of various application scenarios, boasting a high durability, corrosion resistance, and wear resistance, among other benefits [15]. Since the development of bimetallic composites, various preparation processes have emerged, such as centrifugal casting, explosive welding, and hot rolling [16,17,18,19]. Selecting the appropriate preparation process and optimizing its process steps and parameters are essential to improving the performance of bimetallic composite materials. However, some key problems need to be solved in the process of the development and application of bimetallic composites. First of all, in the preparation process of bimetallic materials, there exists the problem of the binding between the matrix metal and cladding metal [20]. The advantage of a high binding strength is that it is not easy to cause delamination and peeling in the subsequent processing [21]. Secondly, when two metal components interact, element diffusion may occur, leading to the formation of intermetallic compounds at the joint of the composite materials [22,23]. This could result in a brittle fracture of the materials when subjected to external forces, which would restrict the improvement in the mechanical properties of the bimetallic composites [24]. Therefore, optimizing the process parameters, achieving high-strength metallurgical bonding of the bimetallic composites, and controlling harmful element diffusion are crucial for the development and effective application of bimetallic composites.

SS/CS bimetal composite bolts, which combine the corrosion resistance of the SS cladding on the surface and high strength of the CS base material in the core, are a new type of high-strength and high-corrosion-resistant composite bolt that can fully utilize the advantages of the two metal materials. At the same time, it can significantly save material costs and reduce project costs. Currently, these bolts are being widely used in the fields of marine engineering and petrochemicals. However, in addition to the high strength and high corrosion resistance requirements for bolts in the application fields of marine engineering, there are generally also requirements for a high fatigue strength or fatigue life. However, the study of the fatigue performance of composite bolts is still blank and needs further research, especially with regard to bending fatigue. In the rotating process of the impeller, the connection bolt at the root of the fan blade for the fastening is prone to fatigue due to periodic bending stress. Bending fatigue is an important cause of bolt relaxation, damage, and even fracture. Therefore, it is of great significance to study the bending fatigue strength of bolts for improving the fatigue testing system of these bolts. In this paper, bimetal composite bolts were created using 45 CS core 304 SS cladding. A microstructure analysis was carried out through a metallographic analysis and microhardness testing. Corrosion resistance was conducted through electrochemical testing and salt spray testing. Through the rotating bending fatigue test method, the fatigue performance of the composite bolt was tested and studied. The microstructure of the fatigue fracture was observed and analyzed using a stereomicroscope and scanning electron microscope (SEM). Finally, the fatigue curves of the bolts were plotted.

## 2. Materials and Methods

The primary production process of the composite bolt involved inserting a 45# CS rod (φ20.8 × 1500 mm) into a 304 SS tube (outside diameter 25.4 × 1500 mm, thickness 2.2 mm), which formed a composite blank. The composite blank was then heated to 1250 °C and subjected to six passes of hot rolling before being cold-rolled into a thread shape, resulting in the production of the composite bolt. A schematic of the hot rolling process is shown in Figure 1. The 304/45-CW bolt underwent further treatment, including quenching and tempering. The bolt was placed in a box-type furnace and quenched at a temperature of 850 °C for 60 min, before being immersed in water for cooling. It was then tempered at a temperature of 500 °C for 60 min and cooled in air to room temperature (25 °C).

The M14× 100 mm 35K CS bolts and 304/45-CW bolts were subjected to testing using the CWT-100 tensile testing machine. The machine was slowly loaded at a rate of 2 mm/min until the bolts broke. The maximum surface bending stress value was selected based on the tensile strength. The fatigue test was conducted using the XM-1000 cantilever beam rotary bending fatigue testing machine (JNSG, Shandong, China), with the bolt loaded as shown in Figure 2. The speed of the fatigue test was 4000 r/min. After a fatigue fracture occurred, the weight tray of the fatigue testing machine fell down and hit the stop button. The fatigue cycle number of the bolt was taken as the fatigue life of the bolt under the current maximum surface bending stress. The fracture morphology of the samples was observed using a stereomicroscope (Leica S9E, MLGD, Shanghai, China) and SEM (Sigma300, ZEISS, Shanghai, China).

Corrosion electrochemical tests were carried out using a CHI760E electrochemical workstation in a conventional three-electrode cell, consisting of a reference electrode (saturated calomel electrode), counter electrode (platinum), and working electrode (specimen). Potentiostatic tests were conducted at room temperature in a 3.5 wt.% NaCl solution to accelerate the corrosion.

## 3. Results

### 3.1. Materials and Microstructure

#### 3.1.1. Chemical Composition Analysis

304 SS was chosen as the cladding material due to its excellent wear resistance, corrosion resistance, and processability. As the cladding material was relatively soft compared to 35K CS, 45# CS, with a higher strength than 35K CS, was selected as the core material to ensure the strength of the composite bolt. The chemical compositions of the two materials are listed in Table 1.

#### 3.1.2. Microscopic Inspection of Composite Rods

In the composite rod materials, the firm bonding at the joint and the uniformity of the SS cladding thickness were critical factors. If the joint bonding was not firm or the SS cladding thickness was uneven, the roundness of the bolt could be lost or peel, and detachment could occur during the thread rolling. To ensure the processing quality of the subsequent bolts, it was necessary to test the bonding condition of the composite rod material joint and the uniformity of the SS cladding thickness beforehand.

Figure 3 shows the measurement positions of the joint and SS cladding of the 304/45 composite rod. Microscopic observations showed that the SS cladding and CS core were well bonded after the hot rolling, and no defects such as pores or inclusions were observed. The thickness of the SS cladding in the 3, 6, 9, and 12 o’clock directions of the cross-section of the composite rod material are presented in Table 2, with an average thickness difference of approximately 105 μm, indicating that the thickness of the SS cladding was relatively uniform.

#### 3.1.3. Metallographic Microstructure Analysis

A microscopic structure analysis was conducted on the composite bolts made from 45# CS rods, 304/45-CW bolts, and 304/45-QT bolts. Additionally, a microscopic structure analysis was conducted on the commercial 35K traditional CS bolts used for comparison. Figure 4 shows that the original 45# CS consisted of white reticular ferrite and black blocky pearlite, with a grain size of 10 levels and a slight Widmanstätten structure tendency. In the 304/45-CW bolts, the 45# CS core also consisted of white reticular ferrite and black blocky pearlite, but the grain became coarser, with a grain size of 8 levels. There was a more serious Widmanstätten structure tendency with an overheating phenomenon, which may have been due to the high temperature of the hot-rolling process. After the quenching and tempering treatment, the core structure became tempered sorbite. The structure of the 35K CS bolt consisted of white reticular ferrite and black strip pearlite, with a grain size of 11 levels.

The metallographic structure of the thread is presented in Figure 5. From the metallographic photos of the 304 SS bolts in Figure 5a, the austenite grains were significantly flattened and elongated in a fibrous shape due to the effect of the cold rolling and extrusion deformation of the bolts. These grains were distributed in a linear manner, perpendicular to the direction of the thread formation, especially at the root of the thread. At the same time, carbide particles with a diameter of about 1 μm were precipitated at the austenitic grain boundary.

From Figure 5b, it can be seen that the SS cladding of the 304/45-CW bolts also had the fibrous streamline structure characteristic. The size of the austenite grains was very uneven and there were individual abnormally large austenite grains. This was caused by the excessive high-rolling temperature (1250 °C) of the composite rods, which resulted in the abnormal growth of the SS austenite grains. The grain boundaries of the SS cladding were very clear, indicating that intergranular sensitization had occurred [25]. This phenomenon was caused by the carbon element on the CS side diffusing preferentially along the austenite grain boundaries of the SS cladding during the hot-rolling process. The chromium element in the SS reacted to form chromium carbide precipitation on the grain boundaries, resulting in the formation of a chromium-deficient zone on both sides of the grain boundaries [26]. The grain boundaries near the joint connected to form a network with thicker grain boundaries and more severe sensitization. At the same time, small dispersed particles were observed to be distributed at the joint of the SS cladding, with carbide particle diameters of about 1 μm.

The microstructure of the thread in the 304/45-QT bolts is shown in Figure 5c. The SS cladding exhibited a martensitic structure and featured a fibrous flow pattern. In addition to intergranular diffusion, the carbon elements also underwent intragranular diffusion, and a large number of small dispersed carbides were precipitated inside the grains (as seen in the black SS grain regions at the joint in the figure). It can be observed that some dispersed carbide particles grew in size due to the heating, reaching a size of 2 μm, which caused the degree of the intergranular sensitization of the SS cladding in the 304/45-QT bolts to be higher than that of the 304/45-CW bolts. Therefore, it significantly reduced the corrosion resistance of the SS cladding. It is worth noting that the increase in the carbide particle size in the austenitic grains adjacent to the interface not only resulted in a decrease in the corrosion resistance, but also led to a reduction in the mechanical properties in this area.

#### 3.1.4. Microhardness Test

Figure 6 shows the microhardness measurement positions and values of the threads. It can be observed that the microhardness distribution of the SS cladding of the 304/45-CW bolts was uniform from the surface to the joint, with an average microhardness of 474 HV. This was due to the fibrous streamline structure formed by the effect of the thread cold rolling and extrusion deformation on the austenite grains of the SS cladding, which significantly strengthened the microhardness along the fiber direction. Meanwhile, due to the strong cold deformation hardening mechanism of the SS cladding, the surface microhardness was further strengthened. After the QT treatment, the cold deformation strengthening mechanism of the SS disappeared, and the carbon element in the CS core diffused into the SS cladding, forming a large number of small carbide particles (Cr_23_C_6_) at the interface of the SS cladding, which formed a dispersion-strengthening mechanism and produced a microhardness peak (442 HV) at the interface. However, the microhardness peak was still lower than the average microhardness value of the SS layer of the 304/45-CW bolts, indicating that the cold deformation strengthening made a significant contribution to the increase in the SS microhardness. There was a decarburization layer of about 200 μm in the CS layer and the microhardness of the CS increased with an increase in carbon content.

### 3.2. Fatigue Cycle Counting

#### 3.2.1. Fatigue Cycles under Initial Stress

Prior to conducting the rotary bending fatigue test, a bolt tensile test was conducted to determine the load stress range for the fatigue test. It was carried out on the 35K CS bolts and 304/45-CW bolts (M14× 100 mm, 2.0 mm). The average tensile strengths were 578 MPa and 593 MPa, respectively. As a result, 300 MPa was selected as the initial stress of the maximum surface bending stress to choose the loading stress range. Then, according to the stress and the length of the bending moment, the required weight of the weight was calculated. Finally, the weight was loaded and the rotational bending fatigue test was started after setting the speed.

Table 3 displays the fatigue cycles of the bolts subjected to the maximum surface bending stress of 300 MPa. The data in Table 3 were analyzed using Weibull distribution and the results are shown in Figure 7. The slope parameter b of the bolts and characteristic life parameter *N*_a_ when the failure probability was 63.2% were obtained. It can be seen that the fatigue cycles of the 304/45-CW reached 342,600 times at a 63.2% failure probability, which was significantly higher than the 304/45-QT bolts and 35K CS bolts.

The initiation of the fatigue crack source played a critical role in determining the fatigue life of the metallic materials. In the 304 SS bolts, the fibrous structure present on the surface could considerably enhance the mechanical properties in the direction of the fibers. As the fatigue fracture direction of the bolts typically began at the root of the screw teeth, and the fatigue crack propagation direction was perpendicular to the distribution of the fibrous streamline, the fatigue strength of the bolts could be significantly improved by the distribution of the fibrous texture. Moreover, the SS exhibited a high cold deformation strengthening index, leading to intense cold deformation strengthening effects in the root of the screw tooth, along with a large number of residual compressive stresses [27]. These factors helped to further improve the fatigue strength of the bolt, thereby prolonging the fatigue crack initiation time and significantly enhancing the fatigue cycle life of the 304 SS bolts.

The 304/45-CW bolts also had the characteristics of the fibrous streamline structure of the 304 SS bolts. At the same time, due to the high carbon content of the 45# CS, the carbon element in the core material diffused to the SS cladding during the hot rolling process. Dispersed and fine carbide particles were precipitated on the boundary near the joint of the SS cladding. Due to the uniform and tiny particles, a dispersion-strengthening mechanism was formed here, so the microhardness value was greatly improved. Under the combined action of the two influences, the 304/45-CW bolts had a high fatigue strength. After QT at 500 °C, the deformation hardening and strengthening mechanisms of the SS cladding of the 304/45-CW bolt disappeared, the carbide particles grew up with heating, and the dispersion-strengthening effect decreased. Therefore, the fatigue strength of the tempered composite bolt dropped sharply.

#### 3.2.2. Fatigue Curve Measurement

Based on the fatigue test results under the bending stress of 300 MPa, multiple stress values were added to the 304/45-CW bolts and 304/45-QT bolts to determine the S-N fatigue curves. 

In the S-N curve fitting, Basquin’s power function expression as shown in Formula (1) was used.
*σ^m^N* = *c*
(1)

The log of both sides of Formula (1) was taken to obtain Formula (2).
lg*σ* = *a* + *b*lg*N*
(2)

In Formula (2), *a* = lg*c*/*m*, *b* = −1/*m*. Formula (2) shows that there is a log-linear relationship between stress and life.

There is an approximate linear relationship between the rotational bending fatigue limit *σ*_f_ and ultimate strength *σ*_u_ of metal materials and, according to Formula (3), there is an empirical relationship.
*σ*_f_ = *kσ*_u_
(3)

For ductile materials, *σ*_u_ is the yield strength *σ*_s_ and *k* is the coefficient.

Suppose that the fatigue limit for *N* = 10^3^ is 0.9 *σ*_u_. At the same time, the infinite life corresponding to the fatigue limit *σ*_f_ of metal materials is generally *N* = 10^7^ cycles, so it can be conservatively assumed that the life corresponding to the fatigue limit is 10^6^ cycles.

According to Basquin’s formula, Formulas (4) and (5) are obtained.
(0.9*σ*_u_)*^m^* × 10^3^ = *c*
(4)
*σ*_f_*^m^* × 10^6^ = *c*
(5)

Formulas (6) and (7) can be obtained by combining Formulas (4) and (5).
*m* = 3/(lg0.9 − lg*k*)(6)
*c* = lg^−1^{[6lg0.9 + 3(lg*σ*_u_ − lg*k*)]/(lg0.9 − lg*k*)} (7)

According to Equations (6) and (7), if the basic S-N curve is expressed by Basquin’s formula, then *m* = 12.6, *c* = 9.8 × 10^35^. Thus, *a* = 2.86 and *b* = −0.08. The fitted S-N curves are shown in Figure 8. From the curves, it can be seen that the fatigue strength of the 304/45-CW bolts was close to 240 MPa, but the fatigue strength of the 304/45-QT bolts decreased significantly to 85 MPa.

#### 3.2.3. Macroscopic Morphology Analysis of Fatigue Fracture

The macroscopic photographs of the fatigue fracture surfaces of the bolts under the maximum surface bending stress of 300 MPa are shown in Figure 9. The fracture morphology can be divided into three main areas, which correspond to the crack source, extended region, and transient fault region [28]. The crack initiation location in the fatigue test was mostly at the first thread, where the bolt and nut came into contact, which was generally located at the root of the thread because this was where the bolt was subjected to the maximum bending stress. After the fatigue crack initiation, the fatigue crack gradually developed along the transverse section of the bolt.

In the fatigue crack extended region, the bending stress was relatively low and the fatigue crack extended slowly. The fracture surfaces on both sides of the crack were repeatedly collided and squeezed for a long time, resulting in a brighter area in this region. As shown in Figure 9a,b, multiple bright, striated cracks with obvious water wave patterns can be observed at the edge of the fatigue fracture of the bolt, indicating repeated collision and squeezing processes in this region. The bright area in Figure 9c is less, indicating that the time for slow fatigue crack propagation accounted for a small part of the entire fatigue fracture process. In Figure 9d, the area around the fracture surface of the bolt is all bright, indicating that this was a multi-source fatigue fracture. In the rapid fatigue crack propagation zone, the fracture surface was relatively dark and rough.

The surface of the transient fault region was relatively dark and very rough. At this time, the bending stress was relatively high and the effective bearing area of the bolts was significantly reduced. The fatigue crack propagated rapidly and the number of squeezes on both sides of the fracture surface decreased, causing the bolt to experience a final unstable fracture. Fibrous plastic fractures are shown in Figure 9a,c. Dark and rough shear lips, as shown in Figure 9b,d, can be clearly observed.

#### 3.2.4. SEM Micromorphology Analysis

The microstructures of the fatigue striations on the bolts are shown in Figure 10. Clear, band-like fatigue striations can be observed in the fatigue propagation region. The step distance of the fatigue striations was approximately 1 μm for the 35K CS bolts, 304 SS bolts, and 304/45-CW bolts. For the 304/45-QT bolts, the step distance of the fatigue striations was approximately 4 μm. The results indicate that the 304/45-QT bolts had a lower fatigue life because the fatigue crack propagated a longer distance in the fatigue extended region under the same bending stress.

### 3.3. Corrosion Resistance Tests

#### 3.3.1. Polarization Curve Measurement

The polarization curves of the bolt samples in a 3.5 wt.% NaCl solution are presented in Figure 11. The electrochemical parameters obtained from these polarization curves are summarized in Table 4. It was observed that the self-etching potential of the 304/45-CW bolts was -298 mV with a corrosion current density of 1.44 μA/cm^2^. The corrosion resistance of the 304/45-CW bolts exhibited a slight decrease compared to the 304 SS bolts. The self-etching potential of the 304/45-QT bolts decreased to −458 mV, with an increase in the current density to 7.92 μA/cm^2^, resulting in a significant decline in the corrosion resistance. In the hot-rolling process, there was a gap between CS and SS, which made it challenging to achieve sufficient element diffusion behavior. Additionally, the contact time between the CS and SS in the hot-rolling process was relatively short, leading to less carbon element diffusion from CS to SS, resulting in only a slight sensitization of the SS cladding [29]. Therefore, the SS cladding of the 304/45-CW bolts still exhibited a passable corrosion resistance. The corrosion resistance of the 304/45-QT bolts decreased significantly because, during the quenching and tempering treatment, the CS and SS were closely combined and the holding time was as long as 90 min, which was sufficient to allow the carbon elements in the CS to fully spread to the SS, resulting in a more severe sensitization phenomenon and eventually leading to a sharp decline in the corrosion resistance.

#### 3.3.2. Salt Spray Test

To verify the reliability of the polarization curve results, a salt spray test was conducted on the bolts. The test medium was selected as a 3.5 wt.% NaCl solution and the spray method was set to continuous spray. The experimental chamber temperature was set to 35 °C. After 720 h of neutral salt spray corrosion, the bolts’ surfaces were cleaned and dried, as shown in Figure 12. It can be observed that there were no corrosion traces on the surface of the 304 SS bolts (Figure 12a) and 304/45-CW bolts (Figure 12b). However, the surface of the 304/45-QT bolts exhibited minor brown rust stains, as shown in Figure 12c. It can be seen that the SS cladding of the 304/45-CW bolts maintained a good corrosion resistance.

## 4. Conclusions

The composite bolts were well formed and displayed no discernible defects at the joint. The SS cladding of the 304/45-CW bolts exhibited fibrous, streamline tissue characteristics. The SS cladding of the 304/45-CW bolts was mainly strengthened by cold deformation. The SS cladding had a high microhardness, with an average microhardness of 474 HV;Under the condition of the maximum surface bending stress of 300 MPa, the fatigue cycles of the 304/45-CW reached 342,600 times at a 63.2% failure probability, which was significantly higher than that of the 35K CS bolts. The S-N fatigue curves showed that the fatigue strength of the 304/45-CW bolts was close to 240 MPa, but the fatigue strength of the 304/45-QT bolts decreased significantly to 85 MPa, which was due to the loss of the cold deformation strengthening effect, and the surface microhardness of the composite bolts decreased significantly;The corrosion current density of the 304/45-CW bolts was twice that of a 304 stainless steel bolt. It was a little bit affected by carbon diffusion. After the quenching and tempering treatment, under the dual effect of sufficient carbon diffusion and the sensitization temperature range, the intergranular sensitization phenomenon of the SS cladding was aggravated and the corrosion resistance decreased obviously.

## Figures and Tables

**Figure 1 materials-16-04454-f001:**
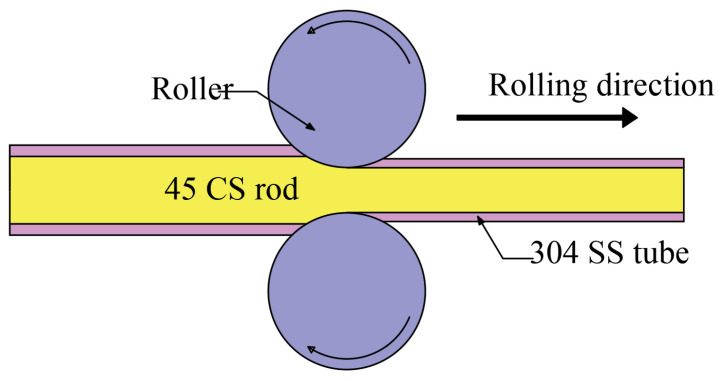
Schematic of the hot rolling process.

**Figure 2 materials-16-04454-f002:**
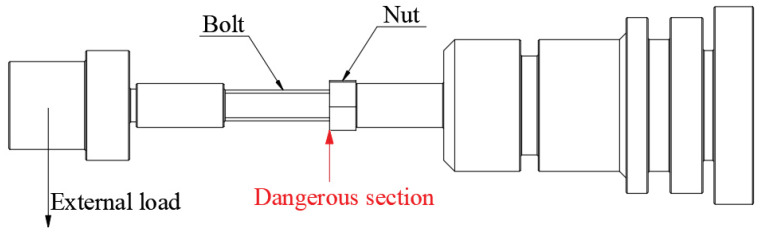
Working diagram of rotary bending fatigue testing machine.

**Figure 3 materials-16-04454-f003:**
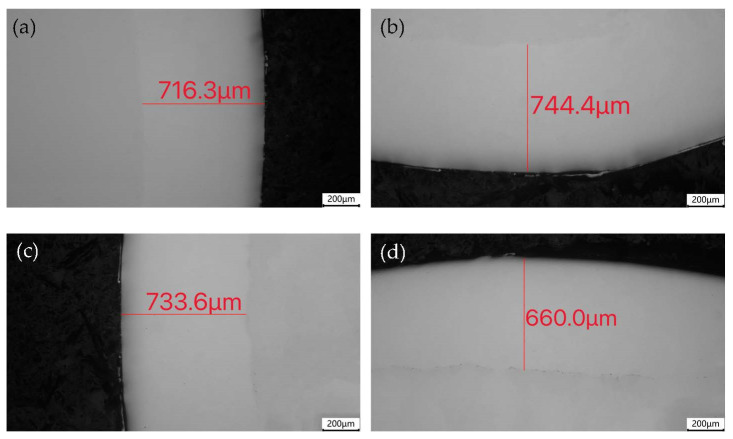
Measurement of the SS cladding thickness in the 304/45 composite rod: (**a**) 3 o’clock; (**b**) 6 o’clock; (**c**) 9 o’clock; and (**d**) 12 o’clock.

**Figure 4 materials-16-04454-f004:**
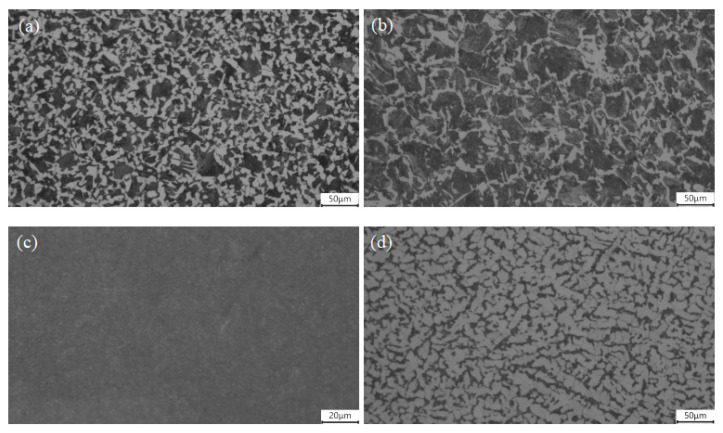
Metallographic structure of core material (**a**) 45# CS rods; (**b**) 304/45-CW bolts; (**c**) 304/45-QT bolts; and (**d**) 35K CS bolts.

**Figure 5 materials-16-04454-f005:**
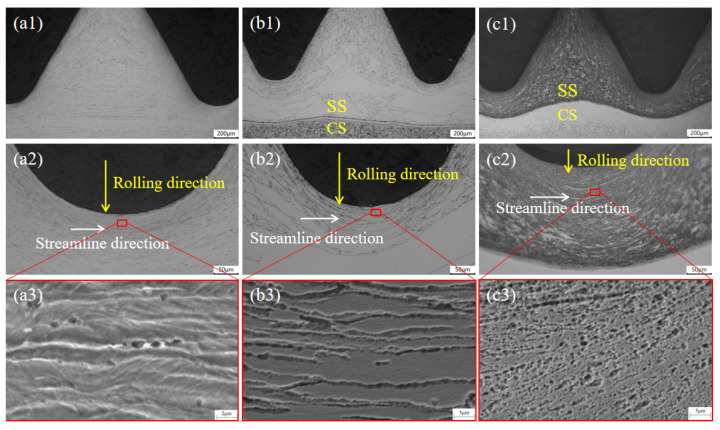
Metallographic structure of thread (**a1**,**a2**) 304 SS bolts; (**b1**,**b2**) 304/45-CW bolts; (**c1**,**c2**) 304/45-QT bolts; and (**a3**–**c3**) partial enlarged images.

**Figure 6 materials-16-04454-f006:**
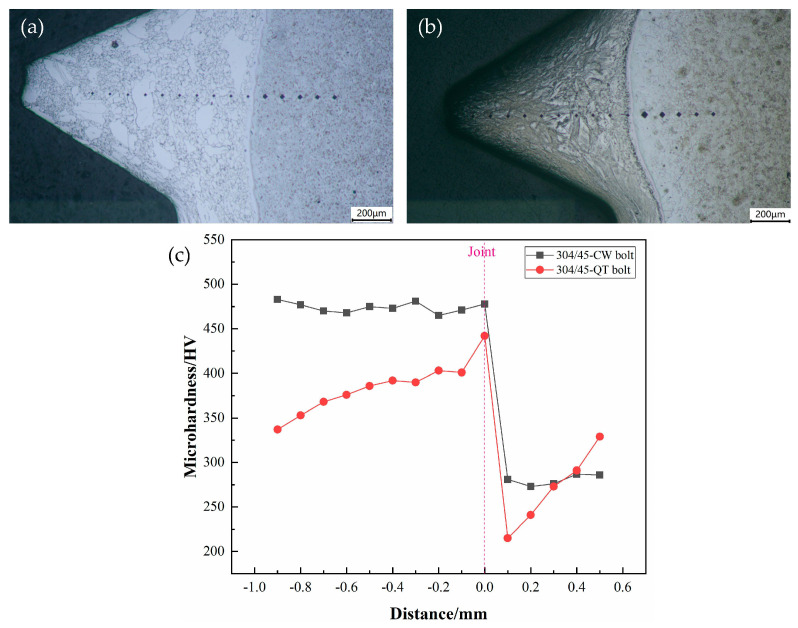
Screw teeth microhardness measurement point and microhardness value of bolts (**a**) 304/45-CW bolts; (**b**) 304/45-QT bolts; and (**c**) microhardness value.

**Figure 7 materials-16-04454-f007:**
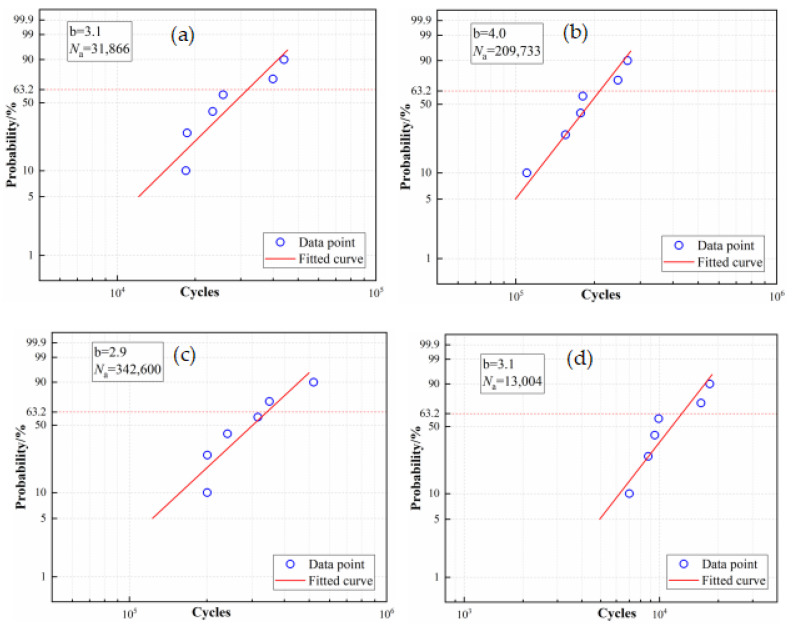
Weibull analysis diagram (**a**) 35K CS bolts; (**b**) 304 SS bolts; (**c**) 304/45-CW bolts; and (**d**) 304/45-QT bolts.

**Figure 8 materials-16-04454-f008:**
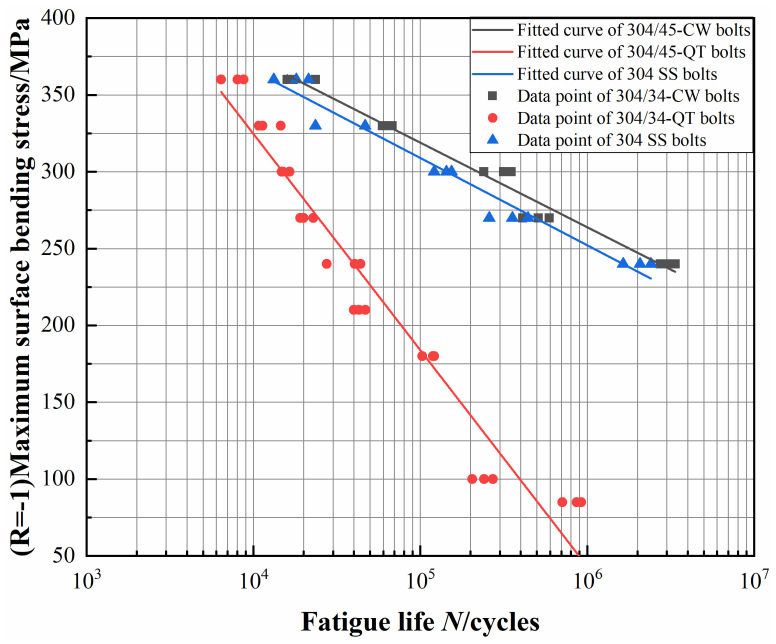
Fatigue curves of 304/45-CW bolts and 304/45-QT bolts.

**Figure 9 materials-16-04454-f009:**
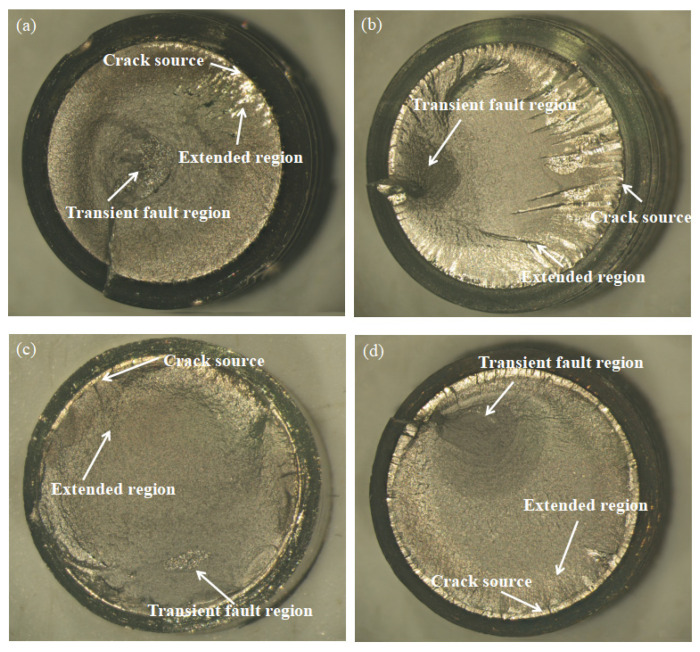
Fatigue fracture morphology of bolts (**a**) 35K CS bolts; (**b**) 304 SS bolts; (**c**) 304/45-CW bolts; and (**d**) 304/45-QT bolts.

**Figure 10 materials-16-04454-f010:**
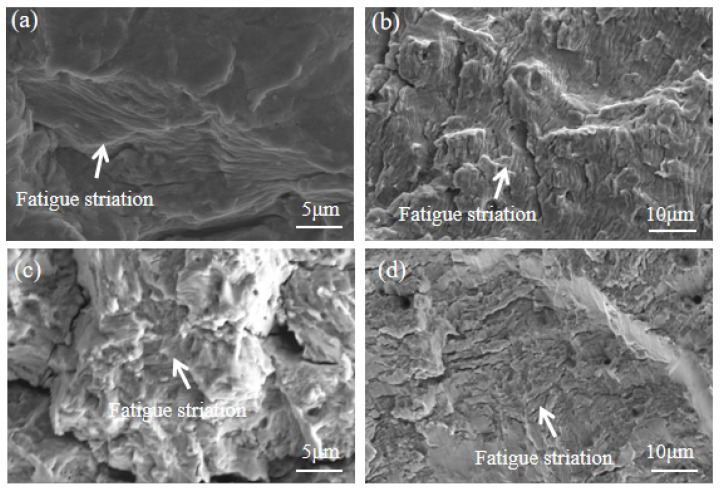
SEM microstructure of bolt fatigue striation under maximum surface bending stress of 300 MPa (**a**) 35K CS bolts; (**b**) 304 SS bolts; (**c**) 304/45-CW bolts; and (**d**) 304/45-QT bolt.

**Figure 11 materials-16-04454-f011:**
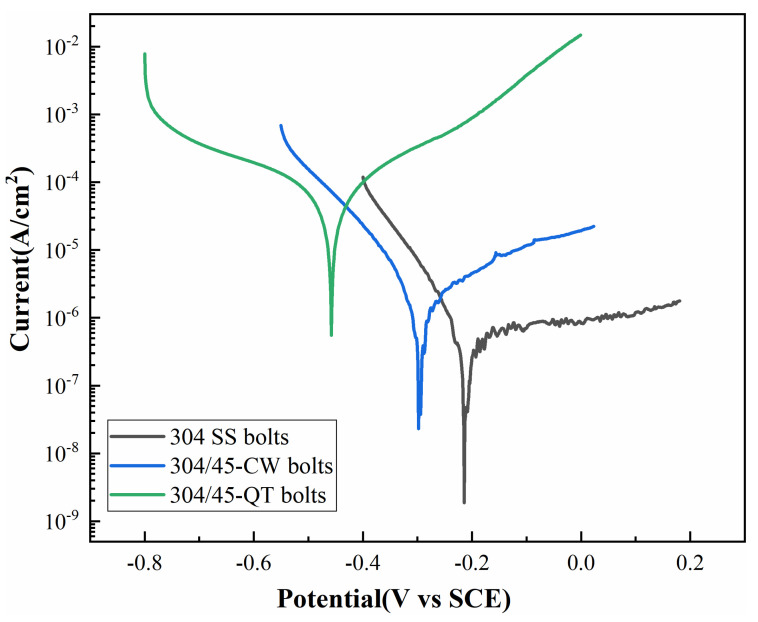
Polarization curves of bolts SS cladding.

**Figure 12 materials-16-04454-f012:**
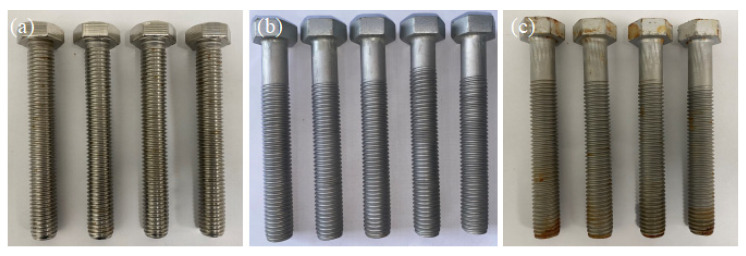
Surface condition of bolts after 720 h salt spray test (**a**) 304 SS bolts; (**b**) 304/45-CW bolts; and (**c**) 304/45-QT bolts.

**Table 1 materials-16-04454-t001:** Chemical composition test results (wt.%).

Material	Cr	Ni	C	Mn	Si	P	S	Fe
304 SS tube	18.18	8.18	0.057	0.73	0.39	0.026	0.006	balance
ASTM type 304 SS	18.0–19.0	9.0–10.0	≤0.07	≤2.00	≤1.00	≤0.045	≤0.030	balance
45 CS rod	0.039	0.016	0.463	0.592	0.228	0.023	0.004	balance
45 CS	≤0.25	≤0.25	0.42–0.50	0.50–0.80	0.17–0.37	≤0.035	≤0.035	balance

**Table 2 materials-16-04454-t002:** Measurement of the SS cladding thickness on the 304/45 composite rods/μm.

Sample	3 o’clock	6 o’clock	9 o’clock	12 o’clock
1	716.3	744.4	733.6	660.0
2	809.3	655.7	510.7	681.7
3	522.0	517.7	502.1	720.6
Average	682.5	640.5	582.1	687.4

**Table 3 materials-16-04454-t003:** Cycles of bending fatigue test cycles of bolts under the maximum surface bending stress of 300 MPa.

Samples	1	2	3	4	5	6
35K CS bolts	18,416	18,632	23,381	25,666	40,021	44,116
304 SS bolts	110,318	155,197	177,310	180,783	246,231	268,016
304/45-CW bolts	200,754	201,041	240,751	316,003	350,198	520,321
304/45-QT bolts	7003	8755	9451	9905	16,301	18,074

**Table 4 materials-16-04454-t004:** Electrochemical parameters of clad layer under different bolts in 3.5 wt.% NaCl solution.

Sample	Self-Etching Potential Ecorr (mV)	Corrosion Current Density Icorr (μA/cm^2^)
304 SS bolts	−214	0.709
304/45-CW bolts	−298	1.44
304/45-QT bolts	−458	7.92

## Data Availability

Not applicable.

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
