# Peer review of "A Study on the Fatigue Performance and Corrosion Resistance of 304/45 Bimetallic Composite Bolts"

_materials, 2023, doi:10.3390/ma16124454_

Round 1

Reviewer 1 Report

The article presents an in-depth study of the mechanical properties of bolts made of a bimetallic composite between AISI 304 and steel 45, with the improved qualities demonstrated by a comparison between standard 35К CS bolts, 45 CS bolts and the bimetallic ones. By means of adequate experimental methods and techniques, the obtained microhardness, microstructure - optical and SEM microscopy, fatigue behavior, fractography and corrosion resistance were evaluated.

I have the following remarks about the content of the article:

1. Line 30 is missing the square brackets for literary source 8.

2. Line 85 The type of equipment with which the tensile test was performed is missing.

3. To be added on fig.1. the place of application of the external load, important dimensions – (such as the distance from the point of application of the load to the dangerous section), as well as to give the speed at which the fatigue test was performed.

4. To clarify the reason why bolts made of material 35K CS are switched to the study of bolts made of material 45 CS. The subsequent transition from material 45 CS to bimetallic composite 304/45CW is well justified.

5. The resolution of the inscriptions in figure 2 to be improved.

6. To indicate the load with which the microhardness values ​​were obtained. Specify anywhere in the text whether hardness or microhardness is specified. In my opinion microhardness should be used.

7. In addition to the appropriate microstructure of the AISI 304 layer, the introduced residual stresses in the surface and subsurface layers determine the fatigue life. Are the authors able to comment on the type and magnitude of the introduced residual stresses in the bolts?

8. It is accepted that fatigue tests for steels are completed by 107 cycles. For what reason were tests not done up to this number of cycles? In addition, in this way a comparison could not be made in terms of fatigue durability for the different combinations.

Reviewer 2 Report

The paper "Study on fatigue performance and corrosion resistance of 

304/45 bimetallic composite bolts" deals with the fatigue and corrosion performance of composite metal bolts. The testing methodology looks suitable for the purpose results. The article, however, needs extensive experimental details. In addition, the analysis needs to be improved.

L 26. there is a pronoun problem; add WHICH before HAVE

L 28. Add AND or OR before petrochemicals

L30. there is an extra “8”

And many more punctuation errors. 

L346. Delete the hyphen in Provin-cial 

Please do a thorough English check

The phrase "Although SS bolts have good corrosion resistance, their strength is low and the price is high [9,10], so they cannot be widely used in applications that require high strength and economy" contradicts itself.

Perhaps the authors want to expand the bimetal composite section regarding fabrication parameters: size,  pressure and add a schematic or a picture of the process.

Please add details of sample preparation and hardness tester in section 2. Also, add details of the tensile tests described in section 3.2.1 (sample dimensions, test speed, machine used).  Same for the electrochemical tests. What potentiostat, electrochemical cell, reference electrode, and the solution concentration are used?

I do not see the contribution of Fig 1. However, a better image of the fatigue setup may help understand how the bolts were loaded.

Please specify which direction Fig 3 shows. Parallel or perpendicular to the rolling direction?

Please enhance the resolution for Fig 5.

The data in Table 3 is worth publishing. However, the analysis needs to be improved. For example, the Weibull distribution represents fatigue tests, but the authors analyze them using the Gaussian distribution.

It is not clear the failure criteria used to stop the fatigue tests. For example, was is sample rupture of crack initiation (by a drop in load ? )

Bolts usually work under axial load. So why did you load them under bending? Although both stresses are normal, their distribution within the cross-sectional area is quite different.

May I recommend the authors present Fig 6 in a log-log scale? For comparison, the fatigue performance of the base material should be added. The authors may want to data fit the results to a fatigue model, such as Basquin or Walker (depending on the load inversion ratio used).

The last conclusion might be improved by adding a quantitative comparison rather than a “FINE” corrosion resistance  (L 336)

The English usage, punctuation, and spelling need a detailed check.

Round 2

Reviewer 2 Report

If possible, the authors should explain (on the paper, perhaps on the introduction) why the bolts were tested under bending rather than axial load.

Other than that, I am satisfied with this version of the manuscript.

Author Response

Point 1: If possible, the authors should explain (on the paper, perhaps on the introduction) why the bolts were tested under bending rather than axial load.

Response 1: Thank you very much for your valuable input. Regarding this issue, my explanation is as follows: I have added an explanation on the introduction(L 70).